# ADAPTIVE STACKED GRAPH FILTER

## ABSTRACT

We study Graph Convolutional Networks (GCN) from the graph signal processing viewpoint by addressing a difference between learning graph filters with fully-connected weights versus trainable polynomial coefficients. We find that by stacking graph filters with learnable polynomial parameters, we can build a highly adaptive and robust vertex classification model. Our treatment here relaxes the low-frequency (or equivalently, high homophily) assumptions in existing vertex classification models, resulting a more ubiquitous solution in terms of spectral properties. Empirically, by using only one hyper-parameter setting, our model achieves strong results on most benchmark datasets across the frequency spectrum.

## 1 INTRODUCTION

The semi-supervised vertex classification problem (Weston et al., 2012; Yang et al., 2016) in attributed graphs has become one of the most fundamental machine learning problems in recent years. This problem is often associated with its most popular recent solution, namely Graph Convolutional Networks (Kipf & Welling, 2017). Since the GCN proposal, there has been a vast amount of research to improve its scalability (Hamilton et al., 2017; Chen et al., 2018; Wu et al., 2019) as well as performance (Liao et al., 2019; Li et al., 2019; Pei et al., 2020).

Existing vertex classification models often (implicitly) assume that the graph has large vertex homophily (Pei et al., 2020), or equivalently, low-frequency property (Li et al., 2019; Wu et al., 2019); see Section 2.1 for *graph frequency*. However, this assumption is not true in general. For instance, let us take the Wisconsin dataset (Table 1), which captures a network of students, faculty, staff, courses, and projects. These categories naturally exhibit different frequency patterns[1]. Connections between people are often low-frequency, while connections between topics and projects are often midrange. This problem becomes apparent as GCN-like models show low accuracies on this dataset; for example, see (Pei et al., 2020; Chen et al., 2020b; Liu et al., 2020).

This paper aims at establishing a GCN model for the vertex classification problem (Definition 1) that does not rely on any frequency assumption. Such a model can be applied to ubiquitous datasets without any hyper-parameter tuning for the graph structure.

**Contributions.** By observing the relation between label frequency and performance of existing GCN-like models, we propose to learn the graph filters coefficients directly rather than learning the MLP part of a GCN-like layer. We use filter stacking to implement a trainable graph filter, which is capable of learning any filter function. Our stacked filter construction with novel learnable filter parameters is easy to implement, sufficiently expressive, and less sensitive to the filters' degree. By using only one hyper-parameter setting, we show that our model is more adaptive than existing work on a wide range of benchmark datasets.

The rest of our paper is organized as follows. Section 2 introduces notations and analytical tools. Section 3 provides insights into the vertex classification problem and motivations to our model's design. Section 4 presents an implementation of our model. Section 5 summarizes related literature with a focus on graph filters and state-of-the-art models. Section 6 compares our model and other existing methods empirically. We also provide additional experimental results in Appendix A.

---

[1]"Frequency" is an equivalent concept to "homophily" and will be explained in Section 2.

## 2 Preliminaries

We consider a simple undirected graph $G = (V, E)$, where $V = \{1, \ldots, n\}$ is a set of $n$ vertices and $E \subseteq V \times V$ is a set of edges. A graph $G$ is called an attributed graph, denoted by $G(X)$, when it is associated with a vertex feature mapping $X : V \mapsto \mathbb{R}^d$, where $d$ is the dimension of the features. We define the following vertex classification problem, also known in the literature as the semi-supervised vertex classification problem (Yang et al., 2016).

**Definition 1** (Vertex Classification Problem). *We are given an attributed graph $G(X)$, a set of training vertices $V_{\text{tr}} \subset V$, training labels $Y_{\text{tr}} : V_{\text{tr}} \to \mathcal{C}$, and label set $\mathcal{C}$. The task is to find a model $h : V \to \mathcal{C}$ using the training data $(V_{\text{tr}}, Y_{\text{tr}})$ that approximates the true labeling function $Y : V \to \mathcal{C}$.*

Let $A$ be the adjacency matrix of the graph $G$, i.e., $A_{i,j} = 1$ if $(i, j) \in E$ and 0 otherwise. Let $d_i = \sum_j A_{ij}$ be the degree of vertex $i \in V$, and let $D = \text{diag}(d_1, \ldots, d_n)$ be the $n \times n$ diagonal matrix of degrees. Let $L = D - A$ be the combinatorial graph Laplacian. Let $\mathcal{L} = D^{-1/2} L D^{-1/2}$ be the symmetric normalized graph Laplacian. We mainly focus on the symmetric normalized graph Laplacian due to its interesting spectral properties: (1) its eigenvalues range from 0 to 2; and (2) the spectral properties can be compared between different graphs (Chung & Graham, 1997). In recent literature, the normalized adjacency matrix with added self-loops, $\tilde{A} = I - \mathcal{L} + c$, is often used as the propagation matrix, where $c$ is some diagonal matrix.

### 2.1 Graph Frequency

Graph signal processing (Shuman et al., 2012) extends "frequency" concepts in the classical signal processing to graphs using the graph Laplacian. Let $\mathcal{L} = U \Lambda U^\top$ be the eigendecomposition of the Laplacian, where $U \in \mathbb{R}^{n \times n}$ is the orthogonal matrix consists of the orthonormal eigenvectors of $\mathcal{L}$ and $\Lambda$ is the diagonal matrix of eigenvalues. Then, we can regard each eigenvector $u_k$ as a "oscillation pattern" and its eigenvalue $\lambda_k$ as the "frequency" of the oscillation. This intuition is supported by the Rayleigh quotient as follows.

$$r(\mathcal{L}, x) \triangleq \frac{x^\top \mathcal{L} x}{x^\top x} = \frac{\sum_{u \sim v} \mathcal{L}_{u,v}(x(u) - x(v))^2}{\sum_{u \in V} x(u)^2}. \tag{1}$$

where $\sum_{u \sim v}$ sums over all unordered pairs for which $u$ and $v$ are adjacent, $x(u)$ denotes the entry of vector $x$ corresponding to vertex $u$, and $\mathcal{L}_{u,v}$ is the $(u, v)$-entry of $\mathcal{L}$. From the definition we see that $r(x)$ is non-negative and $\mathcal{L}$ is positive semi-definite. $r(x)$ is also known as a variational characterization of eigenvalues of $\mathcal{L}$ (Horn & Johnson, 2012, Chapter 4), hence $0 \leq r(x) \leq 2$ for any non-zero real vector $x$. We use the notation $r(x)$ to denote the Rayleigh quotient when the normalized graph Laplacian is clear from context. The Rayleigh quotient $r(x)$ measures how the data $x$ is oscillating. Hence, in this study, we use the term "frequency" and the "Rayleigh quotient" interchangeably. By the definition, the eigenvector $u_i$ has the frequency of $\lambda_i$.

The labeling $y$ of the vertices is low-frequency if the adjacent vertices are more likely to have the same label. This is a common assumption made by the spectral clustering algorithms (Shi & Malik, 2000; Ng et al., 2002; Shaham et al., 2018). Commonly used terms, homophily and heterophily, used in network science, correspond to low-frequency and high-frequency, respectively.

### 2.2 Graph Filtering

In classical signal processing, a given signal is processed by filters in order to remove unwanted interference. Here, we first design a frequency response $f(\lambda)$ of the filter, and then apply the filter to the signal in the sense that each frequency component $\hat{x}(\lambda)$ of the data is modulated as $f(\lambda)\hat{x}(\lambda)$. Graph signal processing extends this concept as follows. Same as in classical signal processing, we design a filter $f(\lambda)$. Then, we represent a given graph signal $x \in \mathbb{R}^{|V|}$ as a linear combination of the eigenvectors as $x = \sum_i x_i u_i$. Then, we modulate each frequency component by $f(\lambda)$ as $x = \sum_i f(\lambda_i) x_i u_i$. An important fact is that this can be done without performing the eigendecomposition explicitly. Let $f(\mathcal{L})$ be the matrix function induced from $f(\lambda)$. Then, the filter is represented by $f(\mathcal{L})x$.

As an extension of signal processing, graph signal processing deals with signals defined on graphs. In definition 1, each column of the feature matrix $X \in \mathbb{R}^{n \times d}$ is a "graph signal". Let $\mathcal{L} = U \Lambda U^\top$ be

the eigendecomposition where $U \in \mathbb{R}^{n \times n}$ consists of orthonormal eigenvectors. Signal $X$ is filtered by function $f$ of the eigenvalues as follow.

$$\bar{X} = U f(\Lambda) U^\top X = f(\mathcal{L}) X \tag{2}$$

In general, different implementations of $f(\mathcal{L})$ lead to different graph convolution models. For instance, GCN and SGC (Wu et al., 2019) are implemented by $f(L) = (I - \mathcal{L} + (D+I)^{-1/2} L (D+I)^{-1/2})^k$, where the constant term stems from the fact that self-loops are added to vertices and $k$ is the filter order. Generally, the underlying principle is to learn or construct the appropriate filter function $f$ such that it transforms $X$ into a more expressive representation. The filter in GCN is called a low-pass filter because it amplifies low-frequency components (Li et al., 2018; NT & Maehara, 2019).

## 3 SPECTRAL PROPERTIES OF FILTERS

Towards building a ubiquitous solution, we take an intermediate step to study the vertex classification problem. Similar to the unsupervised clustering problem, an (implicit) low-frequency assumption is commonly made. However, the semi-supervised vertex classification problem is more involved because vertex labels can have complicated non-local patterns. Table 1 shows three groups of datasets, each with different label frequency ranges. Notably, WebKB datasets (Wisconsin, Cornell, Texas) have mixed label frequencies; some labels have low frequencies while others have midrange frequencies. Therefore, in order to relax the frequency assumptions, we need to learn the filtering function $f(\lambda)$ in a similar way as proposed by Defferrard et al. (2016).

The filtering function $f(\lambda)$ is often approximated using a polynomial of the graph Laplacian as

$$f(\mathcal{L}) \approx \text{poly}(\mathcal{L}) = \sum_{i=0}^{K} \theta_i \mathcal{L}^i. \tag{3}$$

Because polynomials can uniformly approximate any real continuous function on a compact interval (see, e.g., (Brosowski & Deutsch, 1981)), such approximation scheme is well-justified.

Kipf & Welling (2017) derived their GCN formulation as follows. In their equation 5, they approximated a graph filter $g_\theta$ by Chebyshev polynomials $T_k$ as

$$g_\theta * x \approx \sum_{k=0}^{K} \theta_k T_k (D^{-1/2} A D^{-1/2}) x. \tag{4}$$

Then, they took the first two terms and shared the parameters as $\theta_0 = -\theta_1$ to obtain their equation 7:

$$g_\theta * x \approx \theta \left( I_N + D^{-1/2} A D^{-1/2} \right) x \approx \theta \left( 2 I_N - \mathcal{L} \right) \tag{5}$$

Finally, they extended a scalar $\theta$ to a matrix $\Theta$ to accommodate multiple feature dimensions as

$$Z = \tilde{D}^{-1/2} \tilde{A} \tilde{D}^{-1/2} X \Theta \tag{6}$$

Kipf & Welling (2017) claimed that the weight matrix $\Theta$ can learn different filters, and subsequent works (e.g., (Veličković et al., 2018; Spinelli et al., 2020; Chen et al., 2020b)) also learned filters by $\Theta$. However, neither in theory nor practice it is the case (Oono & Suzuki, 2020). As the construction suggest, a GCN layer only represents a filter of the form $f(\lambda) \approx 2 - \lambda$. To properly learn different graph filters, we should learn the multiplying parameters $\theta_0, \theta_1, \ldots, \theta_K$ in equation 3. In the next section, we propose a learning model which directly learns these multiplying parameters.

## 4 MODEL DESCRIPTION

The previous discussion provided several insights: (1) Vertex classification model's frequency is decided by its filter, (2) a mechanism to match the frequencies of data is necessary, and (3) directly learning the polynomial filter's coefficients is more desirable if we do not want to make any frequency assumption. Based on these observations, we implemented an adaptive Stacked Graph Filter (SGF) model. Figure 1 visually describes SGF.

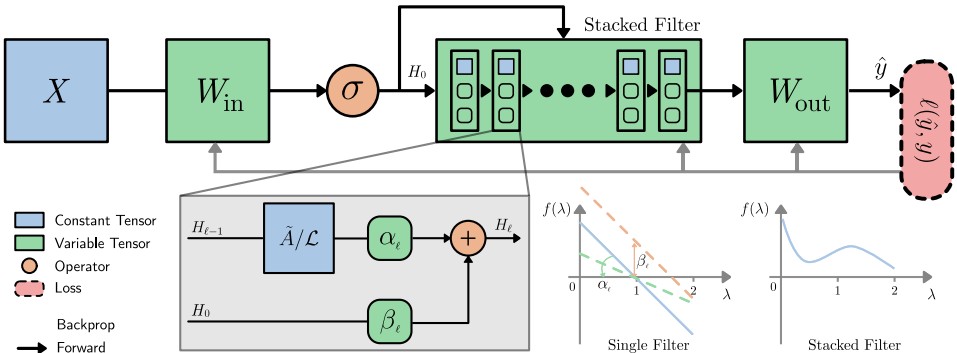

Figure 1: Block description of SGF. $\tilde{A}/\mathcal{L}$ means we can plug either the augmented normalized adjacency matrix or the symmetric normalized Laplacian into this model. In each filter layer, the scalar $\alpha_\ell$ controls the filter's tangent and the scalar $\beta_\ell$ controls the filter's vertical translation.

**Design decisions.**     The novelty of our model is the stacked filter, and we directly learn the filtering function by filter coefficients $\alpha$ and $\beta$, which makes SGF work well universally without frequency hyper-parameters. The deep filter module consists of filters stacked on top of each other with skip-connections to implement the ideas in Proposition 2. Each filter layer has two learnable scalars: $\alpha_\ell$ and $\beta_\ell$ which control the shape of the linear filter (Figure 1). Two learnable linear layers $W_{\mathrm{in}}$ and $W_{\mathrm{out}}$ with a non-linear activation serve as a non-linear classifier (NT & Maehara, 2019).

The input part of our architecture resembles APPNP (Klicpera et al., 2019) in the sense that the input signals (vertex features) are passed through a learning weight, then fed into filtering. The output part of our architecture resembles SGC (Wu et al., 2019) where we learn the vertex labels with filtered signals. This combination naturally takes advantages of both bottom-up (APPNP) and top-down (SGC) approaches. Compared to APPNP and SGC, besides the different in filter learning, our model performs filtering (propagation) on the latent representation and classifies the filtered representation, whereas APPNP propagates the predicted features and SGC classifies the filtered features.

From the spectral filtering viewpoint, our approach is most similar to ChebyNet (Defferrard et al., 2016) since both models aim to learn the filtering polynomial via its coefficients. Chebyshev polynomial basis is often used in signal processing because it provides optimal interpolation points (Cheney, 1966; Hammond et al., 2011). However, since we are learning the coefficients of an unknown polynomial filter, all polynomial bases are equivalent. To demonstrate this point, we implement the Stacked Filter module (Figure 1) using ChebNet's recursive formula in Section 6. We find that Chebyshev polynomial basis approach has similar performance to the stacked approach with one slight caveat on choosing $\lambda_{\max}$. We empirically show this problem by setting the scaling factor $\lambda_{\max} = 1.5$. Note that, as pointed out by Kipf & Welling (2017), such problem can be migrated simply by assuming $\lambda_{\max} = 2$ so all eigenvalues stay in $[-1, 1]$.

Given an instance of Problem 1, let $\sigma$ be an activation function (e.g., ReLU), $\tilde{A} = I - (D + I)^{-1/2} L (D + I)^{-1/2}$ be the augmented adjacency matrix, $\alpha_\ell$ and $\beta_\ell$ be the filter parameters at layer $\ell$, a $K$-layer SGF is given by:

**SGF**: Input $\tilde{A}$

$$H_0 = \sigma(XW_{\mathrm{in}})$$
$$H_\ell = \alpha_\ell \tilde{A} H_{\ell-1} + \beta_\ell H_0, \ \ell = 1\ldots K$$
$$\hat{y} = H_K W_{\mathrm{out}}$$

**SGF**: Input $\mathcal{L}$

$$H_0 = \sigma(XW_{\mathrm{in}})$$
$$H_\ell = \alpha_\ell \mathcal{L} H_{\ell-1} + \beta_\ell H_0, \ \ell = 1\ldots K$$
$$\hat{y} = H_K W_{\mathrm{out}}$$

SGF can be trained with conventional objectives (e.g., negative log-likelihood) to obtain a solution to Problem 1. We present our models using the augmented adjacency matrix to show its similarity to existing literature. However, as noted in Figure 1, we can replace $\tilde{A}$ with $\mathcal{L}$.

The stacked filter is easy to implement. Moreover, it can learn any polynomial of order-$K$ as follows. The closed-form of the stacked filter (Figure 1) is given by

$$\beta_K I + \sum_{i=1}^{K}(\prod_{j=i}^{K} \alpha_j)\beta_{i-1}\mathcal{L}^{K-i+1} \tag{7}$$

where $\beta_0 = 1$. Because each term of equation 7 contains a unique parameter, we obtain the following.

**Proposition 2.** *Any polynomial* $\text{poly}(\mathcal{L})$ *of order* $K$ *can be represented by the form equation 7.*

Note that the same result holds if we replace $\mathcal{L}$ in equation 7 by $\tilde{A}$. In practice, we typically set the initial values of $\alpha_i = 0.5$ and update them via the back-propagation. The learned $\alpha_i$ is then likely to satisfy $|\alpha_i| < 1$, which yields a further property of the stacked filter: it prefers a low-degree filter, because the coefficients of the higher-order terms are higher-order in $\alpha_i$ which vanishes exponentially faster. This advantage is relevant when we compare with a trivial implementation of the polynomial filter that learns $\theta_i$ directly (this approach corresponds to horizontal stacking and ChebyNet (Defferrard et al., 2016)). In Appendix A.1, we compare these two implementations and confirm that the stacked filter is more robust in terms of filter degree than the trivial implementation.

## 5    RELATED WORK

GCN-like models cover a subset of an increasingly large literature on graph-structured data learning with graph neural networks (Gori et al., 2005; Scarselli et al., 2008). In general, vertex classification and graph classification are the two main benchmark problems. The principles for representation learning behind modern graph learning models can also be split into two views: graph propagation/diffusion and graph signal filtering. In this section, we briefly summarize recent advances in the vertex classification problem with a focus on propagation and filtering methods. For a more comprehensive view, readers can refer to review articles by Wu et al. (2020), Grohe (2020), and also recent workshops on graph representation learning[2].

**Feature Propagation.**  Feature propagation/message-passing and graph signal filtering are two equivalent views on graph representation learning (Defferrard et al., 2016; Kipf & Welling, 2017). From the viewpoint of feature propagation (Scarselli et al., 2008; Gilmer et al., 2017), researchers focus on novel ways to propagate and aggregate vertex features to their neighbors. Klicpera et al. (2019) proposed PPNP and APPNP models, which propagate the hidden representation of vertices. More importantly, they pioneered in the decoupling of the graph part (propagation) and the classifier part (prediction). Abu-El-Haija et al. (2019) also proposed to use skip-connections to distinguish between 1-hop and 2-hop neighbors. Zeng et al. (2020) later proposed GraphSAINT to aggregate features from random subgraphs to further improve their model's expressivity. Pei et al. (2020) proposed a more involved geometric aggregation scheme named Geom-GCN to address weaknesses of GCN-like models. Most notably, they discussed the relation between *network homophily* and GCN's performance, which is similar to label frequency $r(Y)$ in Table 1. Spinelli et al. (2020) introduced an adaptive model named AP-GCN, in which each vertex can learn the number of "hops" to propagate its feature via a trainable halting probability. Similar to our discussion in Section 3, they still use a fully-connected layer to implement the halting criteria, which controls feature propagation. AP-GCN's architecture resembles horizontal stacking of graph filters where they learn coefficients $\theta$ directly. However their construction only allows for binary coefficients[3]. We later show that full horizontal stacking models (more expressive than AP-GCN) is less stable in terms of polynomial order than our approach (Appendix A.1). More recently, Liu et al. (2020) continued to address the difficulty of low homophily datasets and proposed a non-local aggregation based on 1D convolution and the attention mechanism, which has a "reconnecting" effect to increase homophily.

**Graph Filtering.** GCN-like models can also be viewed as graph signal filters where vertex feature vectors are signals and graph structure defines graph Fourier bases (Shuman et al., 2012; Defferrard et al., 2016; Li et al., 2018; Wu et al., 2019). This graph signal processing view addresses label efficiency (Li et al., 2019) and provides an analogue for understanding graph signal processing using

---

[2]See, e.g., `https://grlplus.github.io/`

[3]In the manuscript, they showed a construction using coefficients of graph Laplacian, but the actual implementation used GCNConv (which is $I - \mathcal{L} + c$) from pytorch-geometric.

traditional signal processing techniques. For example, the Lanczos algorithm is applied in learning graph filters by Liao et al. (2019). Bianchi et al. (2019) applies the ARMA filter to graph neural networks. Similar to (Klicpera et al., 2019), Wu et al. (2019) and NT & Maehara (2019) also follow the decoupling principle but in a reversed way (filter-then-classify). (Chen et al., 2020b) built a deep GCN named GCNII which holds the current best results for original splits of Cora, Citeseer, and Pubmed. They further showed that their model can estimate any filter function *with an assumption that the fully-connected layers can learn filter coefficients* (Chen et al., 2020b, Proof of Theorem 2).

## 6 EXPERIMENTAL RESULTS

We conduct experiments on benchmark and synthetic data to empirically evaluate our proposed models. First, we compare our models with several existing models in terms of average classification accuracy. Our experimental results show that our single model can perform well across all frequency ranges. Second, we plot the learned filter functions of our model to show that our model can learn the frequency range from the data — such visualization is difficult in existing works as the models' filters are fixed before the training process.

### 6.1 DATASETS

We use three groups of datasets corresponding to three types of label frequency (low, midrange, high). The first group is low-frequency labeled data, which consists of citation networks: Cora, Citeseer, Pubmed (Sen et al., 2008); and co-purchase networks Amazon-Photo, Amazon-Computer (Shchur et al., 2018). The second group is network datasets with midrange label frequency (close to 1): Wisconsin, Cornell, Texas (Pei et al., 2020); and Chameleon (Rozemberczki et al., 2019). The last group consists of a synthetic dataset with high label frequency (close to 2). For the Biparite dataset, we generate a connected bipartite graph on 2,000 vertices (1,000 on each part) with an edge density of 0.025. We then use the bipartite parts as binary vertex labels. Table 1 gives an overview of these datasets; see Appendix B.3 for more detail.

Table 1: Overview of graph datasets, divided to three frequency groups

| DATASETS | $|V|$ | $|E|$ | $d$ | $|\mathcal{C}|$ | $r(Y)$ | $r(X)$ | Type |
|---|---|---|---|---|---|---|---|
| *Cora* | 2,708 | 5,278 | 1,433 | 7 | $0.23 \pm 0.04$ | $0.91 \pm 0.10$ | Citation |
| *Citeseer* | 3,327 | 4,676 | 3,703 | 6 | $0.27 \pm 0.03$ | $0.81 \pm 0.19$ | Citation |
| *Pubmed* | 19,717 | 44,327 | 500 | 3 | $0.55 \pm 0.02$ | $0.87 \pm 0.07$ | Citation |
| *Amz-Photo* | 7,487 | 119,043 | 745 | 8 | $0.25 \pm 0.04$ | $0.82 \pm 0.04$ | Co-purchase |
| *Amz-Computer* | 13,381 | 245,778 | 767 | 10 | $0.27 \pm 0.05$ | $0.83 \pm 0.04$ | Co-purchase |
| *Wisconsin* | 251 | 450 | 1703 | 5 | $0.87 \pm 0.08$ | $0.89 \pm 0.23$ | Web |
| *Cornell* | 183 | 277 | 1703 | 5 | $0.86 \pm 0.11$ | $0.86 \pm 0.32$ | Web |
| *Texas* | 183 | 279 | 1703 | 5 | $0.98 \pm 0.03$ | $0.84 \pm 0.32$ | Web |
| *Chameleon* | 2,277 | 31,371 | 2325 | 5 | $0.81 \pm 0.05$ | $0.99 \pm 0.01$ | Wikipedia |
| *Bipartite* | 2,000 | 50,182 | 50 | 2 | $2.0 \pm 0.00$ | $1.0 \pm 0.00$ | Synthetic |

### 6.2 VERTEX CLASSIFICATION

We compare our method with some of the best models in the current literature. Two layers MLP (our model without graph filters), GCN (Kipf & Welling, 2017), SGC (Wu et al., 2019), and APPNP (Klicpera et al., 2019) are used as a baseline. Geom-GCN-(I,P,S) (Pei et al., 2020), JKNet+DE (Xu et al., 2018; Rong et al., 2019), and GCNII (Chen et al., 2020a) are currently among the best models. We implement the Chebyshev polynomial filter as in (Defferrard et al., 2016) and set $\lambda_{\max} = 1.5$. The *Literature* section of Table 2 and 3 shows the best results found in the literature where these models are set at the recommended hyper-parameters and recommended variants for each dataset. In our experiment, we fix the graph-related hyper-parameters of each model and report the classification results. Our model contains 16 layers of stacked filters ($\tilde{A}$) and has 64 hidden dimensions. Learning rate is set at 0.01, weight decay is $5e \times 10^{-4}$, and dropout rate for linear

layers is 0.7. From an intuition that the filter should discover the required frequency pattern before the linear layers, we set the learning rate of linear layers to be one-fourth of the main learning rate. This experimental setup shows that SGF can adapt to the label frequency without setting specific hyper-parameters. In Table 2, SGF performs comparably with the current state-of-the-art. On the other hand, in Table 3, SGF is not only better than others in our experiments but also surpassing the best results in the literature. Note that we also the *exact same SGF model across all experiments*.

Table 2: Vertex classification accuracy for low-frequency datasets

| METHODS | DATASETS | | | | |
|---|---|---|---|---|---|
| | Cora | Citeseer | Pubmed | Photo | Computer |
| *Our experiments* (Average over 10 runs of stratified 0.6/0.2/0.2 splits) | | | | | |
| MLP | $75.01 \pm 1.33$ | $73.24 \pm 1.28$ | $83.56 \pm 0.44$ | $85.05 \pm 1.62$ | $80.42 \pm 0.73$ |
| SGC ($k = 2$) | $87.15 \pm 1.57$ | $75.00 \pm 0.93$ | $87.97 \pm 0.35$ | $93.67 \pm 0.68$ | $90.87 \pm 0.43$ |
| APPNP ($\alpha = 0.2$) | $88.07 \pm 1.32$ | $76.71 \pm 0.88$ | $88.21 \pm 0.37$ | $94.70 \pm 0.50$ | $91.16 \pm 0.44$ |
| GCNII $(0.5, 0.5)$ | $86.21 \pm 1.40$ | $76.86 \pm 1.29$ | $89.77 \pm 0.52$ | $92.57 \pm 0.61$ | $88.71 \pm 0.55$ |
| SGF-Cheby ($\lambda_{max} = 2.0$) | $\mathbf{88.42 \pm 1.60}$ | $76.85 \pm 1.01$ | $87.74 \pm 0.37$ | $91.26 \pm 1.76$ | $89.71 \pm 0.55$ |
| SGF-Cheby ($\lambda_{max} = 1.5$) | $30.05 \pm 0.60$ | $21.11 \pm 0.03$ | $41.72 \pm 2.99$ | $26.79 \pm 1.82$ | $36.99 \pm 0.03$ |
| **SGF** | $\mathbf{88.97 \pm 1.21}$ | $\mathbf{77.58 \pm 1.11}$ | $\mathbf{90.12 \pm 0.40}$ | $\mathbf{95.58 \pm 0.55}$ | $\mathbf{92.15 \pm 0.41}$ |
| *Literature* (Best result among their variants) | | | | | |
| GCN | 85.77 | 73.68 | 88.13 | (not avail.) | (not avail.) |
| GAT | 86.37 | 74.32 | 87.62 | (not avail.) | (not avail.) |
| Geom-GCN | 85.27 | **77.99** | 90.05 | (not avail.) | (not avail.) |
| APPNP | 87.87 | 76.53 | 89.40 | (not avail.) | (not avail.) |
| JKNet+DE | 87.46 | 75.96 | 89.45 | (not avail.) | (not avail.) |
| GCNII | **88.49** | 77.13 | **90.30** | (not avail.) | (not avail.) |

Table 3: Vertex classification accuracy for midrange and high frequency datasets

| METHODS | DATASETS | | | | |
|---|---|---|---|---|---|
| | Wisconsin | Cornell | Texas | Chameleon | Bipartite |
| *Our experiments* (Average over 10 runs of stratified 0.6/0.2/0.2 splits) | | | | | |
| MLP | $83.72 \pm 3.40$ | $80.13 \pm 4.59$ | $\mathbf{80.30 \pm 5.55}$ | $45.63 \pm 1.88$ | $48.34 \pm 1.67$ |
| SGC ($k = 2$) | $56.27 \pm 6.79$ | $53.37 \pm 5.41$ | $51.49 \pm 6.75$ | $26.51 \pm 2.44$ | $48.07 \pm 1.47$ |
| APPNP ($\alpha = 0.2$) | $71.02 \pm 5.98$ | $74.55 \pm 4.49$ | $66.95 \pm 6.02$ | $54.58 \pm 1.67$ | $50.89 \pm 1.08$ |
| GCNII $(0.5, 0.5)$ | $71.57 \pm 5.13$ | $74.47 \pm 5.42$ | $73.78 \pm 6.72$ | $55.81 \pm 1.55$ | $49.70 \pm 1.75$ |
| SGF-Cheby ($\lambda_{max} = 2.0$) | $76.28 \pm 4.23$ | $69.32 \pm 5.67$ | $77.59 \pm 4.36$ | $\mathbf{70.16 \pm 2.08}$ | $\mathbf{100.0 \pm 0.00}$ |
| SGF-Cheby ($\lambda_{max} = 1.5$) | $52.34 \pm 6.11$ | $59.25 \pm 3.14$ | $62.22 \pm 5.43$ | $28.71 \pm 3.19$ | $\mathbf{100.0 \pm 0.00}$ |
| **SGF** | $\mathbf{87.06 \pm 4.66}$ | $\mathbf{82.45 \pm 6.19}$ | $\mathbf{80.56 \pm 5.63}$ | $58.77 \pm 1.90$ | $\mathbf{100.0 \pm 0.00}$ |
| *Literature* (Best results among their variants) | | | | | |
| GCN | 45.88 | 52.70 | 52.16 | 28.18 | (not avail.) |
| GAT | 49.41 | 54.32 | 58.38 | 42.93 | (not avail.) |
| Geom-GCN | 64.12 | 60.81 | 67.57 | 60.90 | (not avail.) |
| APPNP | 69.02 | 73.51 | 65.41 | 54.30 | (not avail.) |
| JKNet+DE | 50.59 | 61.08 | 57.30 | 62.08 | (not avail.) |
| GCNII | **81.57** | **76.49** | **77.84** | **62.48** | (not avail.) |

Results in Table 3 also suggest that the ability to adapt of the state of the art model GCNII is sensitive to its parameters $\alpha$ and $\theta$. In our experiment, we fix the $\theta$ parameter to 0.5 for all datasets, while in their manuscript the recommended values are around 1.5 depending on the dataset. With the recommended hyper-parameters, GCNII can achieve the average accuracy of $81.57\%$ on Wisconsin data. However, its performance dropped around $3 \sim 10\%$ with different $\theta$ values. This comparison highlights our model's ability to adapt to a wider range of datasets without any graph-related hyper-parameters.

The Chebyshev polynomial basis performs comparably to the staking implementation as we discussed in the previous sections. The value $\lambda_{max} = 1.5$ is choosen because the typical maximum eigenvalue of real-world networks are often at this value. However, in practice, one should set $\lambda_{max} = 2$ as

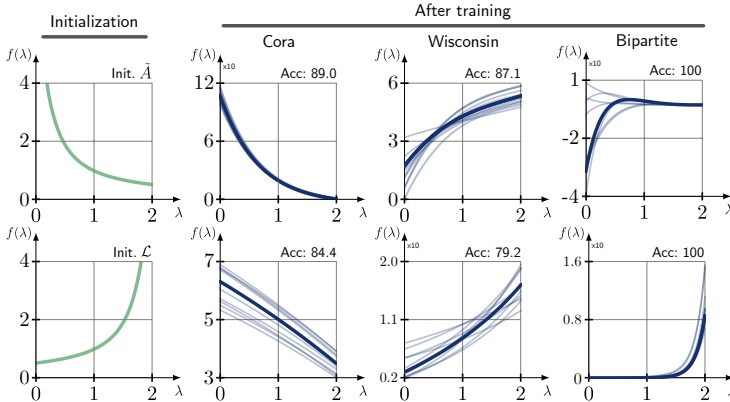

Figure 2: Learned filtering functions $f(\lambda)$ on three datasets corresponding to three frequency ranges. Each row shows the learning results for each initialization. Lightened lines represent the learned filtering functions of 10 different runs. The average accuracy is shown on the top right corner.

discussed by Kipf & Welling (2017). Our experiments here intent to highlight the potential numerical instability problem due to the arbitarily large leading coefficient of the Chebyshev polynomial basis. Since for vertex classification any polynomial basis is equivalent, numerical stable ones like our implementation of SGF is certainly more preferable in practice.

### 6.3 FILTER VISUALIZATION

Another advantage of our model is the ability to visualize the filter function using an inversion of Proposition 2. The first row of Figure 2 shows the filtering functions at initialization and after training when input is the normalized augmented adjacency matrix. The second row shows the results when the input is the normalized Laplacian matrix. These two cases can be interpreted as starting with a low-pass filter ($\tilde{A}$) or starting with a high-pass filter ($\mathcal{L}$). Figure 2 clearly shows that our method can learn the suitable filtering shapes from data regardless of the initialization. We expect the visualization here can be used as an effective exploratory tool and baseline method for future graph data.

### 6.4 ADAPTIVITY TO STRUCTURAL NOISE

Recently, Fox & Rajamanickam (2019) raised a problem regarding structural robustness of a graph neural network for graph classification. Zügner et al. (2018) posed a similar problem related to adversarial attack on graphs by perturbations of vertex feature or graph structure for the vertex classification setting (Dai et al., 2018; Bojchevski & Günnemann, 2019; Zügner & Günnemann, 2019). Here, we evaluate the robustness of the models against the structural noise, where we perturb a fraction of edges while preserving the degree sequence[4]. This structural noise collapses the relation between the features and the graph structure; hence, it makes the dataset to have the midrange frequency. This experimental setting shows that adaptive models like ours and GCNII are more robust to structural noise. In the worst-case scenario (90% edges are swapped), the adaptive models are at least as good as an MLP on vertex features. Figure 3 shows vertex classification results at each amount of edge perturbation: from 10% to 90%. APPNP with $\alpha = 0.2$ and SGC with $k = 2$ have similar behavior under structural noise since these models give more weights to filtered features. On the other hand, APPNP with $\alpha = 0.8$ is much more robust to structural noise as it depends more on the vertex features. This result suggests that adaptive models like ours and GCNII can be a good baseline for future graph adversarial attack studies (SGF's advantage here is being much simpler).

### 6.5 DYNAMICS OF $\alpha$'S AND $\beta$'S

In addition to Section 6.3, this section studies the dynamic of $\alpha$ and $\beta$ during training for two representative datasets: Cora (low-frequency) and Wisconsin (mid-frequency). We the value of $\alpha$ and $\beta$ in SGF ($\tilde{A}$) every 20 training epochs and plot the result. Figure 4 shows the values of $\alpha$ and $\beta$ in 16 layers of SGF in top to bottom then left to right order (reshaped to 4 by 4 blocks). For the Cora dataset, we see that the over-smoothing effect is quickly migrated as the $\alpha$'s automatically go to zero with the exception of the last three layers. Similarly, the weights for skip-connections – $\beta$'s – quickly

---

[4]https://en.wikipedia.org/wiki/Degree-preserving_randomization

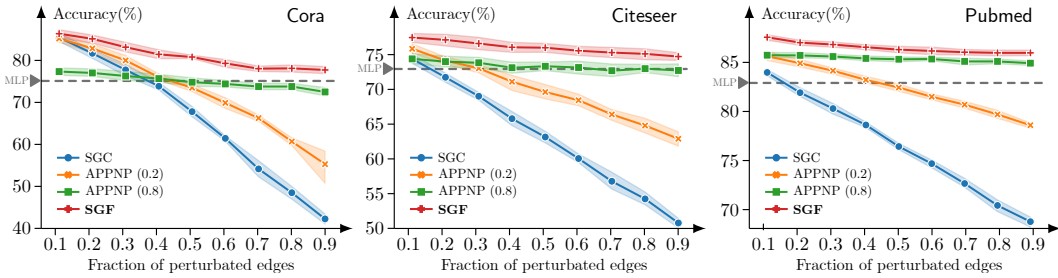

Figure 3: Vertex classification accuracy for each amount of edge perturbation. Since GCNII has similar performance as our model in this setting, we only plot the results for SGF.

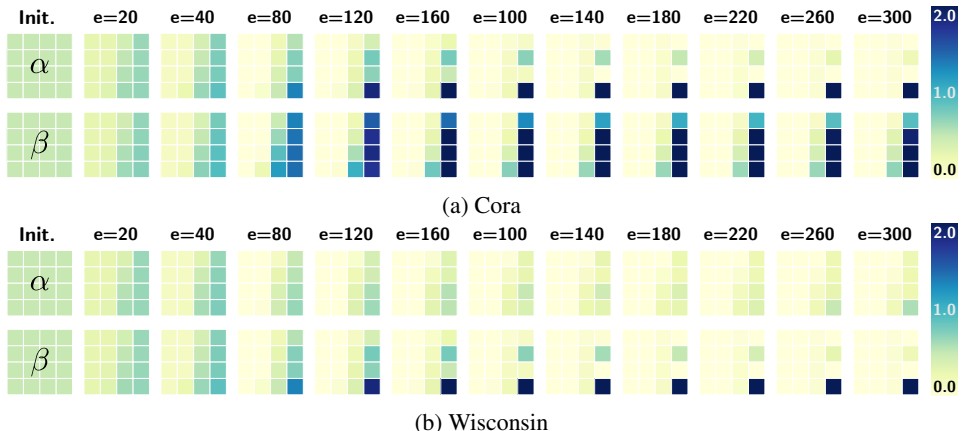

(a) Cora

(b) Wisconsin

Figure 4: Dynamic of $\alpha$'s and $\beta$'s with fixed initialization at 0.5.

go to zero with the exception of few last layers. For the Wisconsin dataset, we can see that there is almost no filtering because all $\alpha$'s go to zero quickly and there is only one active skip-connection in the last layer. This single active skip-connection phenomenon is further confirmed by the experiment on MLP (Table 3) where MLP performed comparably to graph-based models. These results further explained the ability to adapt of our model.

**Additional Experiments.** We provide several other experimental results in Appendix A. Section A.1 discusses the advantages of vertical stacking (SGF) versus a naïve horizontal stacking (learning $\theta$ in equation 3 directly). Section A.2 discusses the difficulty of estimating the frequency range (Rayleigh quotient) of vertex labels when the training set is small. Section A.3 provide additional experiments where $\alpha$'s and $\beta$'s are initialized randomly. We show that our model is still adaptive even with uniform $[-1, 1]$ initialization.

## 7  CONCLUSION

We show that simply by learning the polynomial coefficients rather the linear layers in the formulation of GCN can lead to a highly adaptive vertex classification model. Our experiment shows that by using only one setting, SGF is comparable with all current state-of-the-art methods. Furthermore, SGF can also adapt to structural noise extremely well, promising a robust model in practice. Since our objective is to relax the frequency assumption, one could expect our model will perform weakly when number of training data is limited. Because the estimation of label frequency becomes difficult with a small number of data (Appendix A.2), designing a learning model that is both adaptive and data-efficient is an exciting challenge. We believe an unbiased estimation (Proposition 4) with a more involved filter learning scheme is needed to address this problem in the future.

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

# A  EXTRA EXPERIMENTAL RESULTS

## A.1  VERTICAL AND HORIZONTAL STACKING

Horizontal stacking is equivalent to learning $\theta$'s in Equation 3 directly instead of stacking them vertically. In parallel to our work, Chien et al. (2020) explored the horizontal stacking idea with the pagerank matrix instead of the Laplacian matrix discussed here. We find that both vertical and horizontal can learn degree K polynomial, but vertical stacking naturally robust to the high order terms. Horizontally stacked filter even loses its ability to adapt when learning order 64 polynomials. Table 4 shows a comparison between vertical stacking (SGF) and horizontal stacking. We also report the average number of iteration until early stopping and average training time per epoch for the 64 filters case. All hyper-parameters are the same as in Table 2 and 3. Figure 5 gives an example of 4 layers stacking to clarify the difference between horizontal and vertical.

Table 4: Vertex classification accuracy comparison between horizontal and vertical stacking

| DATASETS | NUMBER OF STACKED FILTERS | | | #Iteration | Time |
|---|---|---|---|---|---|
| | 16 | 32 | 64 | | |
| *SGF* (Average over 10 runs of stratified 0.6/0.2/0.2 splits) | | | | | |
| Cora | $88.97 \pm 1.21$ | $88.70 \pm 1.29$ | $88.75 \pm 1.07$ | 234.5 | 115.4 ms |
| Pubmed | $90.12 \pm 0.40$ | $89.93 \pm 0.55$ | $88.34 \pm 0.67$ | 357.9 | 205.1 ms |
| Wisconsin | $87.06 \pm 4.66$ | $85.51 \pm 4.84$ | $86.17 \pm 4.41$ | 502.5 | 98.6 ms |
| Cornell | $82.45 \pm 6.19$ | $80.55 \pm 6.58$ | $81.14 \pm 4.50$ | 615.7 | 98.7 ms |
| *SGF-Horizontal* (Average over 10 runs of stratified 0.6/0.2/0.2 splits) | | | | | |
| Cora | $88.34 \pm 1.70$ | $88.48 \pm 1.41$ | $88.08 \pm 1.65$ | 765.9 | 107.1 ms |
| Pubmed | $87.38 \pm 0.38$ | $87.27 \pm 0.40$ | $87.10 \pm 0.37$ | 603.8 | 130.6 ms |
| Wisconsin | $84.03 \pm 2.39$ | $78.42 \pm 6.70$ | $60.19 \pm 4.96$ | 1046.5 | 122.1 ms |
| Cornell | $64.95 \pm 6.02$ | $56.84 \pm 5.97$ | $56.83 \pm 6.08$ | 666.8 | 81.5 ms |

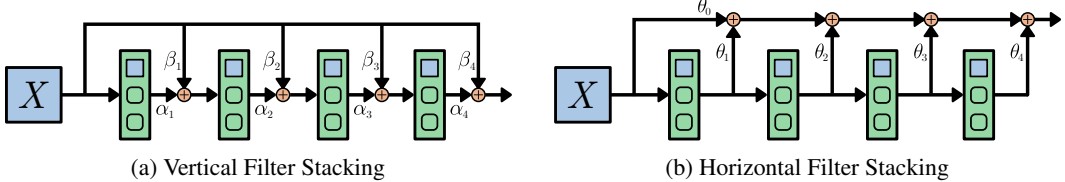

(a) Vertical Filter Stacking                 (b) Horizontal Filter Stacking

Figure 5: Block diagram example of order-4 stacked filters. Both of these models can learn order-4 polynomial as the filtering function.

## A.2  RAYLEIGH QUOTIENT ESTIMATION FROM TRAINING DATA

To obtain an accurate classification solution, the frequency of the model's output must be close to the frequency of the true labels as follows.

**Proposition 3.** *Let $\hat{y}, y \in \mathbb{R}^N$ be unit length vectors whose signs of entries indicate predicted labels and true labels for vertices in graph G. Let $\mathcal{L} \in \mathbb{R}^{n \times n}$ be the symmetric normalized graph Laplacian of graph G. Suppose the graph frequency gap is at least $\delta$: $|r(\hat{y}) - r(y)| = |\hat{y}^\top \mathcal{L} \hat{y} - y^\top \mathcal{L} y| \geq \delta$. Then we have:*

$$||\hat{y} - y||_2^2 \geq \delta/4 \qquad (8)$$

This proposition explains that a model designed for a specific frequency range (e.g., GCN, SGC, GAT, APPNP, etc for low-frequency range) gives a poor performance on the other frequency ranges. This proposition also leads us a method to seek a model (i.e., a filter) whose output matches the frequency of the true labels. Because the true label frequency is unknown in practice, we must estimate this quantity from the training data. Below, we discuss the difficulty of this estimation.

A naïve strategy of estimating the frequency is to compute Rayleigh quotient on the training set. However, training features $X$ and training labels $y_n$ often have Rayleigh quotient close to 1 (as shown in Table 1 for $r(X)$), and Figure 7 (Appendix) shows the results when we compute the Rayleigh quotient of labels based on training data. This means that a naïve strategy yields undesirable results and we need some involved process of estimating the frequency.

If we can assume that (1) Training vertices are sampled i.i.d., and (2) we know the number of vertices in the whole graph ($N = |V|$), we can obtain an unbiased estimation of the frequency of the true labels as follows.

**Proposition 4.** *Let $p$ be the proportion of vertices will be used as training data, $q$ be the proportion of label $y$, $N$ be the total number of vertices in the graph, $\mathcal{L}_n$ be the symmetric normalized Laplacian of the subgraph induced by the training vertices, and $y_n$ be the training labels. Assuming the training set is obtained by sampling the vertices i.i.d. with probability $p$, we can estimate the Rayleigh quotient of true labels by*

$$\mathbb{E}(r(y_n)) = 4N^{-1}p^{-2} \left( y_n^\top \mathcal{L}_n y_n - (1-p) y_n^\top \mathrm{diag}(\mathcal{L}_n) y_n \right) \tag{9}$$

Figure 6 shows an unbiased estimation results using Proposition 4. Unfortunately, at 10% training ratio, the observed variances are high across datasets; thus, we conclude that estimating the label frequency is generally difficult, especially for small training data.

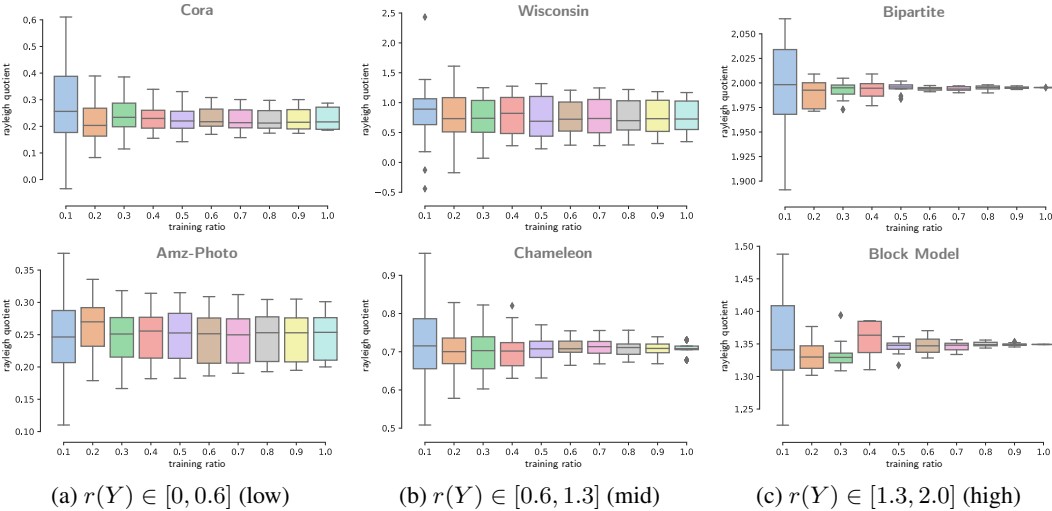

(a) $r(Y) \in [0, 0.6]$ (low)        (b) $r(Y) \in [0.6, 1.3]$ (mid)        (c) $r(Y) \in [1.3, 2.0]$ (high)

Figure 6: Box plot of estimated Rayleigh quotient (frequency) by training ratio. For each training ratio, we randomly sample 10 training sets and compute Rayleigh quotients using equation (9).

Thus far, we have shown that estimating label's frequency given limited training data is difficult even with an unbiased estimator. The high data efficiency of GCN-like models could be contributed to the fact that they already assume the labels are low frequency. Without such assumption, we need more data in order to correctly estimate the frequency patterns.

## A.3   RANDOM INITIALIZATION

While the main content of our paper showed the results for $\alpha$ and $\beta$ initialized at 0.5, our results generally hold even if we initialize them randomly. Table 5 demonstrates this claim by showing our model's performance with $\alpha$ and $\beta$ initialized randomly. SGF (0.5) is the setting showed in the main part of our paper. SGF (U[-1,1]) initializes $\alpha$ and $\beta$ using a uniform distribution in [-1,1].

Both Table 5 and Figure 8 show that our model behaves similar to the fixed initialization at 0.5. It is worthwhile to mention that Figure 8a and 8b show SGF initialized randomly at the same seed but converged to two different solutions. The accuracies for these two particular cases are 89.7% for Cora nd 92.0% for Wisconsin. This result and the filter visualization in Section 6.3 refute the argument that our model is also biased toward "low-frequency".

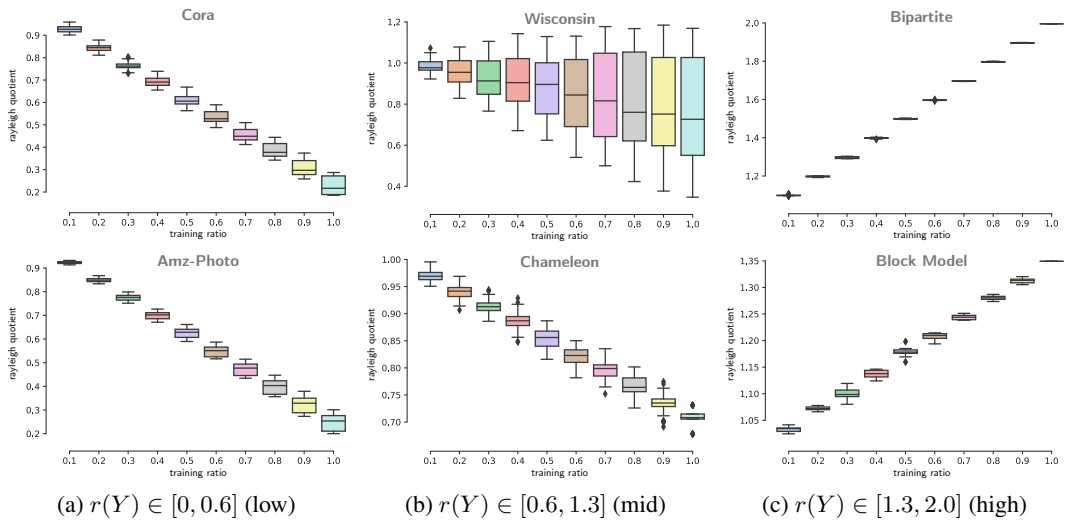

(a) $r(Y) \in [0, 0.6]$ (low)      (b) $r(Y) \in [0.6, 1.3]$ (mid)      (c) $r(Y) \in [1.3, 2.0]$ (high)

Figure 7: Box plot of estimated Rayleigh quotient (frequency) by training ratio. For each training ratio, we randomly sample 10 training sets and compute Rayleigh quotients using equation (1).

Table 5: Test accuracy when $\alpha$ and $\beta$ are initialized randomly

| METHODS | DATASETS | | | | |
|---|---|---|---|---|---|
| | Cora | Citeseer | Pubmed | Photo | Computer |
| SGF (0.5) | $88.97 \pm 1.21$ | $77.58 \pm 1.11$ | $90.12 \pm 0.40$ | $95.58 \pm 0.55$ | $92.15 \pm 0.41$ |
| SGF (U[-1,1]) | $88.47 \pm 1.40$ | $77.50 \pm 1.88$ | $88.23 \pm 1.12$ | $92.23 \pm 0.53$ | $87.15 \pm 3.63$ |
| | Wisconsin | Cornell | Texas | Chameleon | Bipartite |
| SGF (0.5) | $87.06 \pm 4.66$ | $82.45 \pm 6.19$ | $80.56 \pm 5.63$ | $58.77 \pm 1.90$ | $100.0 \pm 0.00$ |
| SGF (U[-1,1]) | $88.66 \pm 3.40$ | $79.13 \pm 1.60$ | $79.67 \pm 3.62$ | $57.83 \pm 2.47$ | $100.0 \pm 0.00$ |

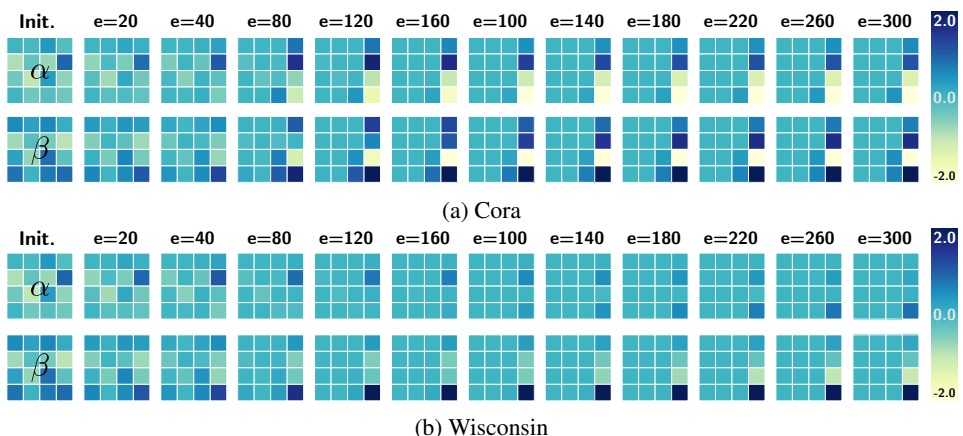

(a) Cora

(b) Wisconsin

Figure 8: Dynamic of $\alpha$'s and $\beta$'s with random initialization (seed 0).

## B  EXPERIMENTAL DETAILS

### B.1  SOURCE CODE

The source code is provided in `src.zip`. The instruction to install Python environment and running examples can be found in `README.md`. All results in this paper are obtained using a single machine with an RTX Titan GPU (24GB). We also confirm the results on CPU and another machine with a GeForce 1080Ti GPU (11GB). The provided source code works on both CPU and GPU.

### B.2  EVALUATION PROCEDURE

For each dataset and each run, the following training procedure is implemented: Split, use train and validation vertices to estimate Rayleigh quotient; train the model with train set and choose the hyper-parameters using validation set, the hyper-parameters are dropout rate, learning rate, and number of layers; save the model every time best validation accuracy is reached; load the best model on validation set to evaluate on test set. Search set for each hyper-parameters:

- Dropout rate: $\{0.4, 0.5, 0.6, \mathbf{0.7}, 0.8\}$
- Weight decay : $\{1e-2, 1e-3, \mathbf{5e\text{-}4}, 1e-4, 5e-5\}$
- Learing rate: $\{0.001, \mathbf{0.01}, 0.02, 0.1\}$
- Number of layers: $\{4, 8, \mathbf{16}, 32, 64\}$

We use the hyper-parameters in bold text to report the result in the main part of our paper.

### B.3  DATA SOURCE

Our datasets are obtained from the pytorch-geometric repository and the node-classification-dataset repository on GitHub. These datasets are "re-packed" with `pickle` and stored in `src/data`. The original URLs are:

- `https://github.com/rusty1s/pytorch_geometric`
- `https://github.com/ryutamatsuno/node-classification-dataset`

**Citation Networks.**  Cora (ML), Citeseer, and Pubmed (Sen et al., 2008) are the set of three most commonly used networks for benchmarking vertex classification models. Vertices in these graphs represent papers, and each of them has a bag-of-word vector indicating the content of the paper. Edges are citations between papers. Originally these edges are directed, but they are converted to undirected edges in the trade-off between information loss and efficiency of methods.

**WebKB.**  WebKB dataset is a collection of university websites collected by CMU[5]. As we mentioned in previous sections, this dataset is special because it contains many different types of vertices that have mixed frequencies. We use the Wisconsin, Cornel, Texas subsets of this dataset.

**Wikipedia.**  The Chameleon dataset belongs to a collection of Wikipedia pages where edges are references and vertex labels indicate the internet traffic. Originally this dataset was created for the vertex regression task, but here, we follow Pei et al. (2020) to split the traffic amount into 5 categories.

The synthetic dataset is generated using NetworkX library and labeled by its bipartite parts. The features are generated randomly with Gaussian $\mathcal{N}(0, 1)$.

### B.4  OTHER METHODS

Other methods are obtained from their respective repository on GitHub. The following are parameter settings for the "Our experiment" section of Table 2 and 3. Since each dataset has a different hyper-parameter values, we follow the author's recommendation for the hyper-parameter not mentioned here. We confirm the results with the recommended hyper-parameters and report them in the "Literature" sections.

---

[5]`http://www.cs.cmu.edu/afs/cs.cmu.edu/project/theo-11/www/wwkb`

- GCNII: $\theta = 0.5$, $\alpha = 0.5$.
- SGC: $k = 2$, lr $= 0.01$, wd $= 5 \times 10^{-4}$, dropout $= 0.7$.
- APPNP: $K = 2$, $\alpha = 0.2$ and $0.8$, lr $= 0.02$, wd $= 5 \times 10^{-4}$, dropout $= 0.7$.
- SGF-Cheby (our implementaion): $\lambda_{\max} = \{1.5, 2.0\}$, K = 16 and other hyper-parameters are the same as SGF.

