# OpenReview forum: "Adaptive Stacked Graph Filter"
_ICLR.cc/2021/Conference — Reject_

### Official Review · AnonReviewer1 · 2020-10-20
**Interesting idea but have weaknesses on both novelty and experiments**

**Rating:** 4
**Confidence:** 5

**Review:**

Summary:
The authors proposed to learn the polynomial graph filter in their model. It can be viewed as adaptively learning the propagation part of APPNP and follows by a linear transformation (in features). They show the proposed model can perform well on both homophilic and heterophilic graphs.

Pros:
1.	The idea of adaptively learn the polynomial filter seems correct and reasonable.
2.	Results on filter visualization and structural noise are interesting.

Cons:
1.	The proposed methodology is not novel. A very similar idea has been proposed previously. (See Detail comments)
2.	Problems of over-smoothing.
3.	Results on experiment section (Table 2 and 3) are questionable.

Detail comments:

While the proposed idea of adaptively learning the polynomial graph filter is interesting, it has been proposed previously in not only the GNN literature [2] but also PageRank based methods [3]. Both of them proposed the idea of adaptively learn the polynomial graph filter, or equivalently the generalized PageRank weights. Hence, I do not think the current paper is completely novel. Nevertheless, the proposed methodology seems to be the correct answer for GNN to adapt to both homophilic and heterophilic graphs. One problem of the current proposed method is that why it can avoid over-smoothing when stacking many layers? The authors use a fixed initialization $\alpha = 0.5$ which is the same as APPNP so at least at the very beginning it won’t suffer from over-smoothing. However, it is unclear how will the coefficients behave during and after training. Also, it is not clear how to initialize $\beta$ in the model. Furthermore, if the proposed model can indeed adaptively learn the good polynomial graph filter, why doesn’t the random initialization work? Does that mean the implicit bias of the specific initialization proposed in the paper is necessary? If that is the case, then I do not see why the claim of “adaptive learning” is correct since it is actually sensitive to the initialization.

Beside the methodology and novelty, I also find the experiment section questionable. Firstly, since the main theme of the paper is learning the polynomial filter, the authors should at least compare their method with ChebNet (GCN-Cheby)[5] which also use polynomial filter. Note that in both [4] and [2], they all show that ChebNet can better adapt to heterophilic graphs compare to GCN and GAT.

On the other hand, according to Appendix B.4, the authors use $K=2$ (propagation step) for APPNP. This is *NOT* the suggested hyperparameter reported in [1] ($K=10$). Note that the authors of [1] even show that if we choose a larger $K\geq 10$, the performance can be slightly improved on Cora, Citeseer and PubMed. In contrast, SGF use $K=16$ which is not a fair comparison to APPNP. There should be a experiment that compares APPNP with SGF under the same $K$.

Finally, the authors claim the performance of most baseline methods are found in the literature. However, this is also problematic to me. Note that in the original GCN and GAT paper, the date split is much sparse then the $0.6/0.2/0.2$ split proposed by the authors. Also, in the Geom-GCN paper they do test their model on Chameleon in the split $0.6/0.2/0.2$. Why is it stated as not available? Even if we assume all the problem above can be well explained, the improvement of the proposed model seems not statistically significant. For example, on Wisconsin, Cornell and Texas, although SGF has the highest accuracy in average, the standard deviation is very large. MLP is within 1 standard deviation. Please report the confidence interval to show that the gain of SGF is indeed statistically significant. On the other hand, SGF is worse than not only SGC but also GCNII by a large margin on Chameleon. If SGF can indeed learn the near-optimal polynomial filter, then why this is the case? At last, in the original Geom-GCN paper, they also have the Actor dataset. I think it would be great if the authors can put this result at least in the Appendix.

Besides these weaknesses, I still find the paper well written. Also, the experiment on filter visualization and structural noise are quite interesting. I believe the paper can be greatly improved if all the concerns above can be addressed.

Minor comments:

In page 2, the authors state that the normalized adjacency matrix with added self-loops is $\tilde{A}= I-D^{-1/2}A D^{-1/2} + c$, where $c$ is some diagonal matrix. This is incorrect. Note that when we add self-loops, the degree matrix $D$ has to changed accordingly. Please see the correct expression in [1] for example.

In page 2, the Rayleigh quotient $r(\mathcal{L},x)$ is defined with two input arguments but later the authors ignore $\mathcal{L}$. While it is clear from the context, the notation is not rigorous.

In page 1 introduction section, the authors mention that the model does not need to tune the hyper-parameters. However, in the same page contribution section, the authors mention that they use one hyper-parameter setting. According to their experiment section, I think what they mean is the previous. It would be great to clarify the ambiguity here.

Reference:

[1] “Predict then Propagate: Graph Neural Networks meet Personalized PageRank,” Klicpera et al., ICLR 2018.

[2] “Adaptive Universal Generalized PageRank Graph Neural Network,” Chien et al., arXiv:2006.07988.

[3] “Adaptive diffusions for scalable learning over graphs,” Berberidis et al., In Mining and Learning with Graphs Workshop @ ACM KDD 2018, pp. 1, 8 2018.

[4] “Generalizing Graph Neural Networks Beyond Homophily,” Zhu et al., NeurIPS 2020. (arXiv:2006.11468)

[5] “Convolutional neural networkson graphs with fast localized spectral filtering,” Defferrard et al., NeurIPS 2016.

---

> ### Author Response · Authors · 2020-11-22
> **Some clarifications on our manuscript**
>
> Let us first summarize and assign numbers to your comments:
>
> 1. Not novel (pointed out one arxiv and one adaptive diffusion in 2018)
> 2. Compare with ChebNet.
> 3. K=12 for APPNP (K=2 is not fair!)
> 4. Add Actor dataset, review Chameleon.
> 5. How to init alpha, beta; how do they behave during training?
> 6. A = I - L + c is wrong?
> 7. r(L,x) = r(x).
> 8. Introduction has repetitive part about hyper-parameter setting.
> 9. Why can it avoid over-smoothing?
>
> ====
>
>
> (1) (2018) is feature less and (2020) is very similar to our Horizontal stacking case and the main difference is that they use the pagerank matrix while we provide arguments and experimental results for a general polynomial basis of both A and L.
>
> (2) It is not "better" than ChebNet, we included the detailed discussion in page 4 and compared with ChebNet (our implementation).
> Both our model and ChebNet learn filters; however, the implementation of ChebNet is sensitive to the choice of \lambda_{max} in the sense that if \lambda_{max} > 1 then the leading polynomial coefficient will be arbitrarily large as more layers are added. Also, ChebNet-GCN is clearly unfair, our implementation showed that ChebNet is fully capable of going deeper.
>
> (3) We find that K=12 and K=2 for APPNP does not differ much, so we reported the experimental result for K=2. In the literature result section, it was the reported result for K=12 by GCNII paper.
>
> (4) We added the Actor datasets. For Chameleon, we find that graph parameters (GCNII) doesn't affect the accuracy, but weight decay is the main contributor for higher accuracy. Hence, Chameleon result is not ideal for our case.
>
> (5) We included the behavior of \alpha and \beta in our updated manuscript.
>
> (6) Thank you very much for pointing out the mistake, we fixed the formulation.
>
> (7) We believe r(L,x) = r(x) is quite clear.
>
> (9) It avoids over-smoothing thanks to the parameter \beta. As seen in figure 1, \beta can "lift"each component filter horizontally, which can migrate the over-smoothing effect automatically.
> Another explanation is that over-smoothing is reported for models having no skip connections. Our model has the \beta-skip-connections to deal with the over-smoothing effect.
>
> We would like to express our thanks for your time and your thoughtful comments. It would be great if you can check our updated manuscript and let us know if the updated version clarified some of your concerns. A yes/no answer for each point would be suffice!

---

> > ### Comment · AnonReviewer1 · 2020-11-25
> > **Some quick comments**
> >
> > Thanks for the response. Although I haven't had time to go through all the detail yet, I would like to post some quick comments for the clarifications before the discussion phase end.
> >
> > For (3), as I mentioned previously, the authors of APPNP [1] have shown that at least on Cora, Citeseer and PubMed, using $K\geq 10$ is better then $K=2$. Please see the Figure 4 in [1]. I'm not sure why there's an inconsistency between your experiment versus the results in [1].
> >
> > For (4), I am not sure I understand the statement "graph parameters (GCNII) doesn't affect the accuracy" and why Chameleon is not ideal for SGF. If the weight decay is indeed the main contributor, I wonder if SGF can be greatly improved using the better weight decay hyperparameter. If not, my concern is still there.
> >
> > For (5), it is good to see how the $\alpha$ and $\beta$ look like. However, my original question is why we have to use a fixed initialization? As you mentioned that SGF can adaptively learn the graph filter, it should not depends on the initialization (that is, the random initialization should also work). Otherwise, it seems to me that the implicit bias offered by the fixed initialization is crucial for SGF to perform well, which makes the conclusion of "adaptive learning" doubtful.

---

> > > ### Author Response · Authors · 2020-11-25
> > > **Thanks for your quick reply**
> > >
> > > >  I'm not sure why there's an inconsistency between your experiment versus the results in [1].
> > >
> > > First, it might be the value of teleportation probability alpha in Figure 4 is 0.1, ours is 0.2. Second, it also could be the splits and number of experiments (100 versus 10 in our case). Nonetheless, our message remains: alpha needs to be tuned in order for APPNP to learn different frequency pattern, no matter number of layers. We also report the literature result in the "literature" section following procedure in GCNII.
> > >
> > > > I wonder if SGF can be greatly improved using the better weight decay hyperparameter.
> > >
> > > For both GCNII and SGF, the result changes greatly with weight decay. Thanks to yours and other reviewers' suggestion, we found that the Chebyshev polynomial basis give a very good result for Chameleon (our updated manuscript). So we think choosing good weight decay can lead to a better solution for SGF. However, as you can see, our main objective is to relax the hyper-parameter tuning, so we show the results as if no hyper-parameters are tuned for SGF. Of course, as another reviewer noted, there is nothing wrong with hyper-parameter tuning.
> > >
> > > > However, my original question is why we have to use a fixed initialization?
> > >
> > > Please see the appendix to see random initialization. Generally, random or fixed initialization for alpha and beta does not matter much. It's just that we want to keep the matter as simple as possible, so we fix the initialization (arbitrarily) at 0.5 for both parameters, one can use any initialization in [-1, 1] as we showed in the Appendix. Thank you again for suggesting this point!
> > >
> > > Thank you again for your comments and your quick response. We do not hope you will change your mind but we hope our response clear some misunderstanding about our work.

---

> > > > ### Author Response · Authors · 2020-11-25
> > > > **Extra on adaptivity**
> > > >
> > > > P/S: In the filter visualization, we show that our model can even take the graph Laplacian as the input and it still can provide a reasonable results on all datasets. You can run our code with the flag `--use_laplacian` to see how it can adapt to different graph matrix. This is another point to show the adaptivity of this simple stacked method because even with fixed initialization, A and L give very different filter shapes (one is low frequency at first and one is high frequency at first - as shown in our filter visualization). Note that there is no change in the learning procedure, just changing the input matrix.

---

### Official Review · AnonReviewer4 · 2020-10-28
**The Review**

**Rating:** 5
**Confidence:** 5

**Review:**

This paper proposes to stack the graph filters with learnable polynomial parameters to construct the new graph neural network model. Generally, this paper is well organized and easy to read. Here are my concerns.

1.Essentially, this paper argues that the approximation of Chebyshev polynomials in GCN can only capture the low-frequency features in the spectral domain, and proposes a more general approximation scheme by stacking the graph filter in the spatial domain.  However, the low-frequency property of GCN is highly related to the localized first-order approximation of graph convolutions. Without this first-order approximation, GCN model can capture the high-frequency information in graphs, e.g, ChebyNet [2] with large enough order K. It's better to add more discussions/comparisons with this kind of GCNs.

Moreover,  my core concern is the superiority of why the proposed polynomial approximation (in Equation 7) is better than the previous Chebyshev approximation from both theoretical and practical justifications.  In graph signal processing, using a polynomial series to approximate the graph filter has been well studied in the literature. As pointed out by [1], Chebyshev polynomial is a good approximator to approximate graph filters. It is better to add more justifications (e.g., numerical analysis) about the proposed approximation scheme.

2.Another concern is the experiment.
Dataset splitting: It seems like that this paper adopts the new splitting plan (stratified 0.6/0.2/0.2 splits) for all datasets. Meanwhile, the paper also reports the best results reported in the literature. However, I think it’s improper to put them in the same table since we can’t make a fair comparison under different data splitting.  Moreover, I would like to see the results of SGF on the public splitting of these datasets.

Hyperprameters: In Appendix B.4, the authors claim that they follow the hyperparameter recommendation in the original paper of baselines. However, it seems that some of the given hyperparameters are not the best hyper-parameters. For example, for Cora, \alpha of GCNII is set to 0.2, while in   Appendix B.4, \alpha=0.5 which inconsistent with the original paper [3].
On the other hand, In Appendix B.2, the authors adopt the random strategy to search the hyperparameters of SGF. Since the authors re-run all the experiments of baselines in the new splits, it’s better to conduct the same hyper-parameter search process for each baseline to ensure a fair comparison.

The filter parameters visualization: From the model construction perspective, since the only difference between SGF and GCNII/APPNP is the trainable filter parameters. Therefore, I’m curious about the value of \alpha and \beta after the training. Could you visualize the value of two parameters in each layer from SGF?

Overall, I think this paper is marginally below the acceptance threshold.

[1] David K. Hammond, Pierre Vandergheynst, and Re ́mi Gribonval. Wavelets on graphs via spectral graph theory. Applied and Computational Harmonic Analysis, 30(2):129–150, 2011.
[2] Defferrard, Michaël, Xavier Bresson, and Pierre Vandergheynst. "Convolutional neural networks on graphs with fast localized spectral filtering." Advances in neural information processing systems. 2016.
[3] Chen, M., Wei, Z., Huang, Z., Ding, B., & Li, Y. (2020). Simple and deep graph convolutional networks. arXiv preprint arXiv:2007.02133.

---

> ### Author Response · Authors · 2020-11-22
> **We updated our manuscript to accommodate your comments**
>
> Let us first summarize and assign numbers to your comments:
>
> 1. Why is it better than ChebNet?
> 2. Justify the "new" splitting plan.
> 3. Justify the hyper-parameter scheme.
> 4. Visualize alpha, beta for each layers.
>
> ====
>
> Missing a more detailed discussion on ChebNet (as pointed out by other reviewers) is indeed our oversight and we agree with your comments on our manuscripts. Here, we would like to clarify some of your concerns:
>
> (1) It is not "better" than ChebNet, we included the detailed discussion in page 4 and compared with ChebNet (our implementation).
> Both our model and ChebNet learn filters; however, the implementation of ChebNet is sensitive to the choice of \lambda_{max} in the sense that if \lambda_{max} > 1 then the leading polynomial coefficient will be arbitrarily large as more layers are added.
>
> (2) We think this is a slight misunderstanding. This splitting plan is not new, it is used by Geom-GCN, GCNII (they called it full-supervised, we just say straightforward what it is). (https://openreview.net/pdf?id=S1e2agrFvS, page 8, paragraph 4).
>
> (3)  While \theta=1.5 is a recommended hyper-parameters for Chameleon and Texas, it is not recommended for other datasets. Actually, GCNII has different “recommended” hyperparameters (\alpha, \theta, number of layers, weight decay) to each dataset. In “Our Experiment” sections, we demonstrate that if we fix a hyperparameter, it does not adapt.
> We emphasize that, as specified by the end of page 6, we reported the best results (with recommended hyper-params by the authors) in the literature sections of Table 2 and 3.
>
> (4) We included the visualization on page 9 and the appendix.
>
> We would like to express our thanks for your time and your thoughtful comments. It would be great if you can check our updated manuscript and let us know if the updated version clarified some of your concerns. A yes/no answer for each point would be suffice!

---

> > ### Comment · AnonReviewer4 · 2020-11-24
> > **The response after rebuttal**
> >
> > Thank you for your response!
> > For point (1), I can understand the explanation, but my core concern is still here. I need more justifications about the superiority of the proposed approximation scheme.
> > For point (2), actually, as far as I know, at least in Cora, Citeseer, and Pubmed,  the data splitting (stratified 0.6/0.2/0.2 splits) of GeomGCN is different from that of GCNII/APPNP/JKNet+DE (Please refer to the appendix in https://openreview.net/pdf?id=Hkx1qkrKPr).  Therefore, I still insist that it’s improper to put them on the same table.  It's better to follow the majority splitting to make a fair comparison.
> > For point (3), yes, you can emphasize that you approach is hyper-parameter insensitive. However, it's unnecessary to enforce other methods to do so to make the improvement look significant.
> > For point (4), I've read the modifications in the paper. It's good and has addressed my concern.
> >
> > Overall, I will keep my mind.

---

> > > ### Author Response · Authors · 2020-11-25
> > > **Thank you for your consideration**
> > >
> > > Thank you for your quick response!
> > >
> > > (1) We understood your standing. Although we want to convince you that these polynomial bases are equivalent, it is our own oversight not to include Chebyshev basis in our first version.
> > >
> > > (2) Thank you for pointing that out, we actually taken the split from GCNII, and we report the literature results following their manuscript. Nonetheless, we will make it clearer in the future submissions. About "enforcing" other methods, we believe we have reported the same result as GCNII and follow their same splitting procedure, so we did provide both views (their optimal results and what happen if hyper-parameters are fixed). The difference here is that we train the filtering parameters.
> > >
> > > Thank you again for your comments and your time. We understand your opinion and while we don't expect you will change your decision, we hope that our rebuttal makes you feel more positive about our work.

---

### Official Review · AnonReviewer2 · 2020-10-29
**The paper proposes a relatively simple formulation for a graph convolutional filter similar to GCNII, that has the advantage of providing useful insights on the characteristic of the considered datasets.**

**Rating:** 5
**Confidence:** 4

**Review:**

Adaptive stacked graph filter
The paper proposes a relatively simple formulation for a graph convolutional filter, that has the advantage of providing useful insights on the characteristic of the considered datasets. Many points of the paper are however not convincing in the present form, mainly regarding the novelty of the proposed formulation.

The paper proposes a graph convolution operator that is inspired by the well-known approximation of a graph filter using polynomials of the graph Laplacian.

Pros:
- The paper proposes a simple filter formulation that allows to study the dependency on the neighborhood radius on different datasets.
- The visualisation of the filters is interesting.
-The reported experimental results are positive, even though in many cases the improvement does not seem significant.

Cons:
-The proposed model is very similar to GCNII: Graph convolution by Kipf and Welling with a single scalar parameters instead of a parameter matrix + skip connections. The main difference with GCNII is the lack of the identity mapping.
In fact, eq. of H^l in page 4 is very similar to eq. 5 in https://arxiv.org/pdf/2007.02133.pdf. Authors should deeply discuss the differences between their proposal and other works in literature, clarifying their novel contribution.

Comments about specific sections follow.
Experimental section:
		-In page 6, authors state that they fix the \theta hyper-parameter of GCNII to 0.5, even though the recommended values are around 1.5.  Can you justify this choice? Also, since you run the experiments on GCNII, it would be interesting to see its performance on the bipartite dataset with \theta = 1.5
		-In Table 3, the results from literature do not report the variance. In general, it seems like the results of the proposed method and baselines are pretty close, and in many cases inside the variance range.

Appendix A:
the horizontal stacking variant is not explained in detail. From the figure it looks like several stacked layers with an aggregation that sums the weighted representation computed at each layer. I don't see why this should be "horizontal". Probably writing down the equations of this model would help.

B.2. While authors state that for each dataset and for each run they select the hyper-parameters using the validation set, later in the same section they state that the results in the main paper are referred to the hyper-parameters in bold. I don't understand how the hyper-parameter selection procedure is adopted.

Minor:
Table 3, Chamaleon dataset. Missing bold on SGC. Texas: MLP is in bold while it shouldn't be
Page 6: "Note that we also the extact" -> we use the

-----REBUTTAL
I acknowledge having checked authors' rebuttal and the revised version of the manuscript

---

> ### Author Response · Authors · 2020-11-22
> **Some clarifications to our experiment settings**
>
> Let us first summarize and assign numbers to your comments:
>
> 1. Too similar to GCNII. Need more discussion.
> 2. Page 6, why GCNII is set at 0.5? Why not 1.5?
> 3. Run at 1.5 for bipartite.
> 4. Literature results doesn't have variance.
> 5. Appendix A: Why "horizontal"? Write down formula.
> 6. Discrepancy of hyperparameters.
> 7. Fix the results in Page 6.
> 8. Improvement doesn't seem significant.
>
> ====
>
> We would like to say that we understand and agree with your suggestions. However, there are some minor points that we believe there is a slight misunderstanding so we hope to clarify them here.
>
> = Misunderstanding points:
>
> (2, 6) While \theta=1.5 is a recommended hyper-parameters for Chameleon and Texas, it is not recommended for other datasets. Actually, GCNII has different “recommended” hyperparameters (\alpha, \theta, number of layers, weight decay) to each dataset. In “Our Experiment” sections, we demonstrate that if we fix a hyperparameter, it does not adapt. We emphasize that, as specified by the end of page 6, we reported the best results (with recommended hyper-params by the authors) in the literature sections of Table 2 and 3. Please visit `https://github.com/chennnM/GCNII/blob/master/full.sh` to see that GCNII uses different hyper-parameter settings for each dataset.
>
> (4) Indeed, the literature results do not have variance. In most other works, authors do not report their variance.
>
> (6) See (2).
>
> (7) The results for SGC is due to a data entry error. We have fixed it in the updated manuscript.
>
> = Rebuttal points
>
> (1) It is true that our model is similar to GCNII, but based on that logic, ours is more similar to APPNP. The main difference to both these modes, as pointed out in page 4, is that we train the polynomial parameters instead of the weight matrices.
>
> (3) We believe by construction of GCNII, setting $\theta=1.5$ do not make it "high-frequency", here is the result for bipartite from original GCNII code:
> ```
> cuda:0 pretrained/6be005effcde4e60abfaaf3a29de50f6.pt
> bipartite
> 0 : 50.25
> 1 : 51.25
> 2 : 49.75
> 3 : 46.50
> 4 : 52.00
> 5 : 49.25
> 6 : 44.50
> 7 : 51.00
> 8 : 44.50
> 9 : 49.00
> Train cost: 29.2409s
> Test acc.:48.80
> ```
>
> (5) We have clarified the horizontal stacking experiment in the updated manuscript.
>
> (8) Indeed some improvement is not statistically significant (i.e., comparable); however our main contribution is that we can obtain such results by using one initialization setting rather than changing hyper-params according to datasets.
>
> We would like to express our thanks for your time and your thoughtful comments. It would be great if you can check our updated manuscript and let us know if the updated version clarified some of your concerns. A yes/no answer for each point would be suffice!

---

> > ### Comment · AnonReviewer2 · 2020-11-23
> > **Comment on author's rebuttal**
> >
> > Dear authors,
> > thank you for spending time to address my comments. I know it is late to post a comment, but I hope you understand that the schedule was very tight also for reviewers (that often are authors as well).
> >
> > From a fast check of your comments and the paper, I don't fully agree with the argument summarized as "our main contribution is that we can obtain such results by using one initialization setting rather than changing hyper-params according to datasets."
> >
> > One thing is to show that a method is not much sensitive to the specific hyper-paramter setting, but there's nothing wrong in selecting the best hyperparameters for each dataset, as long as it is performed in a correct way.
> > On the contrary, claiming that you can obtain good results with the same hyperparameters on many datasets, while true in this case, is necessarily not true in general, given the No Free Lunch theorem.
> > What you want to prove, in my understanding, is that fixed an hyperparameter selection procedure, your proposed method, being less sensitive to hyper-parameters compared to existing alternatives, performs *significantly* better than the baselines.
> >
> > Am I missing something?

---

> > > ### Author Response · Authors · 2020-11-25
> > > **Some comments on NFL**
> > >
> > > Dear reviewer, thank you for having such a quick response to our rebuttal. We know it's hard work to give us comments. We also believe you understand our main point right. In here, we just want to have some discussion.
> > >
> > > > there's nothing wrong in selecting the best hyperparameters for each dataset, as long as it is performed in a correct way.
> > >
> > > This is absolutely right. However, should we learn the parameter that can be learn, rather than searching for them? For example, one simple question is how many GPU needed to come up with these numbers? `https://github.com/chennnM/GCNII/blob/master/full.sh`.  Also, this hyperparameter searching problem was discussed in recent  papers like [Once for All](https://arxiv.org/abs/1908.09791). Hyper-paramter tuning and more advanced neural architecture search could be either wasteful for create a big technical gap across the world. Of course this is not our main reason to develop our model, we just want to provide a different view to hyper-parameter search.
> > >
> > > > NFL
> > >
> > > It is true that all algorithm have the same average performance, so ours performed well here doesn't mean it will perform well on some other datasets. As we mentioned in the conclusion and the Appendix, data efficiency and adaptivity is a trade-off. However, the NFL's setting is also restricted and we cannot apply them without discussing their assumptions. One of the important assumption is that the target function is drawn uniformly at random. In practice, however, (a) we assume that the data has some special property, say, the "essence" (frequency) of the data can be extracted by a polynomial filter with sufficient training samples (Appendix A). (b) We argue that our proposed methods here are suitable for such a property. Note that our assumption (a) is weaker than the assumption in e.g., SGC/GCN that assumes that the essence of the data (low-frequency) is extracted by a pre-designed low-pass filter. Ours works better than the other methods because the data fit well on our assumptions.
> > >
> > > Thank you again for provide such a quick response and thoughtful comments.

---

### Official Review · AnonReviewer3 · 2020-10-31
**Nice paper using stacked graph filters for GCNN architectures, but contribution may fall short**

**Rating:** 5
**Confidence:** 5

**Review:**

SUMMARY:
This paper addresses the problem of vertex classification using a new Graph Convolutional Neural Network (NN) architecture. The linear operator within each of the layers of the GNNN is formed by a polynomial graph filter (i.e., a matrix polynomial of either the adjacency or the Laplacian novelty). Rather than working on the frequency domain, the paper focuses on learning the polynomial coefficients of the filter on the vertex domain. The key novelty is the consideration of a stack architecture for which the polynomial filter is formed by the successive application (i.e., matrix multiplication) of filters of order one. Numerical experiments with real datasets showcase the merits, including superior classification performance, of the proposed architecture.

STRONG POINTS:
The paper is timely and fits nicely the scope of the conference.

The numerical experiments are convincing, offering insights, and demonstrating some of the advantages of the proposed architecture.

The writing is clear, making the paper easy to follow.

WEAK POINTS:
Except for the numerical experiments, I find that the contribution is quite limited. The postulation of GCNN architectures based on polynomial graph filters where the focus is on learning the polynomial coefficients has been studied thoroughly in the literature. In general, the paper does a good job listing relevant works in that area, although some are missing (e.g., Gama - Ribeiro). Some of the existing works look at ARMA structures and recursive order-one filter implementations. I acknowledge that the architecture considered in those papers may not be exactly the same as the one proposed by the authors in this paper. I also appreciate that the application at hand (vertex classification) was not the goal of many of those papers. However, I still feel that the contribution falls short, especially for a top conference such as ICLR. In any case, I am open to change my mind if the authors are able to strengthen their theoretical claims or address my concerns in their rebuttal.

I believe that the title should be changed. GCNN are not mentioned. The current title places the focus on Stacked Graph Filters. My first concern is that, within the linear paradigm (i.e., as polynomials of the adjacency/Laplacian matrix), this type of architectures have already been investigated. More importantly, the paper focuses on NN architectures, so I think it is reasonable to have that on the title.


OVERALL RECOMMENDATION:
Marginal reject. The paper is topical, timely, and nicely written. It addresses a problem of interest and does so with contemporary machine learning tools. The results in real-world datasets are convincing. However, the contribution and novelty are limited, falling short of the average contribution at ICLR.

ADDITIONAL RECOMMENDATIONS:
Being able to obtain additional theoretical results would make the contribution more solid.

Further elaborating on the robustness of the architecture it is another change that would strengthen the manuscript.

---

> ### Author Response · Authors · 2020-11-22
> **Some justifications for our work**
>
> Let us first summarize and assign numbers to your comments:
>
> 1. Polynomial interpolation has been well-studied.
> 2. Lack of theoretical results.
> 3. Title should be changed.
> 4. Elaborate on the robustness of the model.
>
> ====
>
> We agree with your constructive comments and here we just want to justify our motivation for the work.
>
> (1) It is true that polynomial interpolation has been well-studied, and it is our oversight that we did not give representative methods like ChebNet a deeper discussion. We have updated our manuscript (page 4 and Table 2,3) to elaborate this point. Our main argument here is that since we are not finding interpolation points but learning the coefficients, all polynomial bases are equivalent.
>
> (2) Indeed, we are still developing theoretical results from these experiments. However, we hope that reader can see the merit of our paper that "going back to learning polynomial" is beneficial, especially when the training data is abundant (Appendix A discussion).
>
> (3) How do you feel about the new title "Vertex classification - Returns of the polynomials" ?
>
> (4) We will indeed go beyond our discussion in "6.4 ADAPTIVITY TO STRUCTURAL NOISE", but perhaps it can be an entire topic of its own because of the adversarial attack literature.
>
> The new update to our manuscript is marked with dark red text for clarity. We hope this update will clarify some of your questions. We understand that going through manuscripts is time-consuming, so we truly appreciate your comments on our work. If possible, we would like to know if this updated version has clarified your concerns (a simple yes/no for each point would be sufficient).

---

> > ### Comment · AnonReviewer3 · 2020-11-23
> > **Comment on author's responses**
> >
> > Response (1): Not sure if I understand "Our main argument here is that since we are not finding interpolation points but learning the coefficients, all polynomial bases are equivalent." Please elaborate on that.
> >
> > Response (3): new title "Vertex classification - Returns of the polynomials". My original point was that there are many types of graph filters, some of them are linear, some of the are not, some use neural networks, some do not. Given the content of this paper, I think that the terms "graph neural network" or, even better "graph convolutional neural network" should appear in the title. Clearly, this is only a suggestion, the final decision must be made by the authors.
> >
> > Regarding the other issues I raised, I am afraid that addressing them satisfactorily will not be easy, but I need to read the updated paper in detail.

---

> > > ### Author Response · Authors · 2020-11-25
> > > **Reply to the reviewer's responses**
> > >
> > > (1) What we were trying to say is: If we are going to think of the vertex classification problem as polynomial interpolation, which is finding coefficient to a polynomial p(x) = y. Then according to traditional results, Chebyshev polynomials provide minimax-optimal Chebyshev interpolation nodes (set of x_i's to interpolate polynomial coefficient such that p(x_i) = y_i) ; in other words, the roots of Chebyshev polynomial provides interpolation points that miminize the effect of the Runger function. However, in our actual problem here, we are finding polynomial coefficients with already given x_i and y_i, so any polynomial basis is equivalent (solution of one can be translated to another)!
> > >
> > > (3) Thank you. We also believe our overall presentation should be improved.
> > >
> > > We also think that the issues raised by you is difficult to address in such a short amount of time. However, we appreciate your comments and will definitely address them in the future revision of our work. Thank you again for your time!

---

### Decision · Program_Chairs · 2021-01-07
**Final Decision**

**Decision:**

Reject

**Comment:**

The topic covered by the paper is timely, and the way the authors have addressed the problem seems correct. The provided empirical evidence seems to be sufficient to support the main claim of the paper. Presentation is well structured and clear.
Notwithstanding the above merits, the proposed approach seems to confirm other similar proposals presented in the literature, so the contribution of the paper seems to be limited. Although presentation is good, it is not highlighting enough the differences w.r.t. those proposals and the basic approximation result given by Chebyshev polynomials. Especially a better theoretical characterisation w.r.t. to approximation capabilities by Chebyshev polynomials (with no truncation) would have helped to better gain understanding of the merits of the proposed approach. Finally, some of the experimental results do not seem to have a significant  statistical difference w.r.t to the baselines, so it would have helped to have the result of a statistical test.